# The Precipitation Inferred from Soil Moisture (PrISM) Near Real-Time Rainfall Product: Evaluation and Comparison

Thierry Pellarin [1,*], Carlos Román-Cascón [1,2], Christian Baron [3], Rajat Bindlish [4], Luca Brocca [5], Pierre Camberlin [6], Diego Fernández-Prieto [7], Yann H. Kerr [8], Christian Massari [5], Geremy Panthou [1], Benoit Perrimond [1], Nathalie Philippon [1] and Guillaume Quantin [1]

[1] CNRS, IRD, Univ. Grenoble Alpes, Grenoble INP, IGE, F-38000 Grenoble, France; carlosromancascon@ucm.es (C.R.-C.); geremy.panthou@univ-grenoble-alpes.fr (G.P.); benoit.perrimond@univ-grenoble-alpes.fr (B.P.); nathalie.philippon@univ-grenoble-alpes.fr (N.P.); guillaume.quantin@ird.fr (G.Q.)
[2] Laboratoire d'Aérologie, Université Toulouse Paul Sabatier, CNRS, F-31400 Toulouse, France
[3] CIRAD UMR TETIS, Maison de la Télédétection, 500 rue J.F. Breton, F-34093 Montpellier, France; christian.baron@cirad.fr
[4] NASA Goddard Space Flight Center, Greenbelt, MD 20771, USA; rajat.bindlish@nasa.gov
[5] Research Institute for Geo-Hydrological Protection, Via Madonna Alta 126, 06128 Perugia, Italy; luca.brocca@irpi.cnr.it (L.B.); christian.massari@irpi.cnr.it (C.M.)
[6] Centre de Recherches de Climatologie/Biogéosciences, UMR 6282 CNRS, Université Bourgogne Franche-Comté, 21000 Dijon, France; pierre.camberlin@u-bourgogne.fr
[7] EO Science, Applications and Climate Department, Largo Galileo Galilei, 1, 00044 Frascati, Italy; Diego.Fernandez@esa.int
[8] CESBIO (CNRS/UPS/IRD/CNES), 18 av. Edouard Belin, bpi 2801, CEDEX 9, 31401 Toulouse, France; Yann.Kerr@cesbio.cnes.fr
* Correspondence: thierry.pellarin@univ-grenoble-alpes.fr

**Abstract:** Near real-time precipitation is essential to many applications. In Africa, the lack of dense rain-gauge networks and ground weather radars makes the use of satellite precipitation products unavoidable. Despite major progresses in estimating precipitation rate from remote sensing measurements over the past decades, satellite precipitation products still suffer from quantitative uncertainties and biases compared to ground data. Consequently, almost all precipitation products are provided in two modes: a real-time mode (also called early-run or raw product) and a corrected mode (also called final-run, adjusted or post-processed product) in which ground precipitation measurements are integrated in algorithms to correct for bias, generally at a monthly timescale. This paper describes a new methodology to provide a near-real-time precipitation product based on satellite precipitation and soil moisture measurements. Recent studies have shown that soil moisture intrinsically contains information on past precipitation and can be used to correct precipitation uncertainties. The PrISM (Precipitation inferred from Soil Moisture) methodology is presented and its performance is assessed for five in situ rainfall measurement networks located in Africa in semi-arid to wet areas: Niger, Benin, Burkina Faso, Central Africa, and East Africa. Results show that the use of SMOS (Soil Moisture and Ocean Salinity) satellite soil moisture measurements in the PrISM algorithm most often improves the real-time satellite precipitation products, and provides results comparable to existing adjusted products, such as TRMM (Tropical Rainfall Measuring Mission), GPCC (Global Precipitation Climatology Centre) and IMERG (Integrated Multi-satellitE Retrievals for GPM), which are available a few weeks or months after their detection.

**Keywords:** precipitation; soil moisture; Africa; satellite rainfall products; comparison

## 1. Introduction

Rainfall is a crucial resource in Africa, where large parts of the population rely on rainfed agriculture. The continent is also known for its vulnerability to rainfall variability that impacts the natural resources (water, vegetation), and subsequently the wellness of populations, in societies where the economy is based mainly on agriculture [1]. Knowledge of rainfall spatio-temporal distribution is essential to various applications such as water-resource and land-use management, agricultural crop yield estimates, flood nowcasting, dam management, ground-water recharge estimates and irrigation demand. Rain-gauges provide the most common and most direct measurement of point precipitation at the surface, therefore they are generally assumed as the most accurate method to measure precipitation. Unfortunately, Africa is a region where the ground-based rain-gauge network is of very low density and operational radar installations are almost non-existent [2]. Furthermore, the gauge networks have been degrading over the last few decades [3].

In this context, satellite-based precipitation products represent an unavoidable alternative for providing precipitation knowledge in Africa. In recent decades, significant progress has been made in satellite precipitation estimation. This progress is mostly due to the introduction of new sensors (e.g., Global Precipitation Measurement Core Observatory satellite), but also to the improved sensor accuracy, and the efficiency of proposed algorithms that take advantage of the many observational data (including multi-channel VIS/IR sensors, and passive microwave). Many studies have been dedicated to the evaluation of these different satellite precipitation products in Africa [4–8]. Without being exhaustive, the main conclusions of these studies can be summarized as follows: (i) the state-of-the-art products perform relatively well at monthly and decadal time steps [4,9,10] with decreasing performances for finer timescales; (ii) most products satisfactorily reproduce the main features of the rainfall regime [4,5]; (iii) real-time products exhibit moderate to high (positive or negative) biases [4] whereas adjusted or post-processed products, in which ground precipitation measurements are included in algorithms to correct bias, show lower biases, (iv) there is a clear need to improve the accuracy of satellite products in the estimation of accumulated rainfall [6,11].

One potential strategy for improving satellite precipitation products is to use soil moisture measurements from satellite microwave sensors. Soil moisture can be seen as trace precipitation, and a knowledge of the temporal and spatial variability of soil moisture could benefit rainfall retrievals from space. Pioneer studies [12–17] exploited this signal to correct existing satellite precipitation products. Later, refs. [18,19] developed the SM2RAIN approach to directly derive a rainfall amount from soil moisture variations exclusively. Since 2015, these approaches were improved and applied at the global scale [20,21] and on different locations in the US [22], in Australia [23], over selected sites [24] and in China [25]. One of the main advantages of these methodologies based on soil moisture measurement is that they can replace, in near-real-time, the scaling procedures of state-of-the-art satellite precipitation and avoid significant latency for dataset availability.

This paper aims to present the latest developments in the PrIMS (Precipitation Inferred from Soil Moisture) algorithm originally developed by [17,24]. The concept of PrISM is to use an existing real-time precipitation product and to correct it using soil moisture information. In addition to the PrISM algorithm, this paper presents an accurate evaluation of its performances for Africa, including a comparison with the performance of ten additional precipitation products at the daily time-scale. In order to improve the evaluation of the product in a region with very low rain-gauge coverage, exclusively local, national and pan-national in situ rain gauge measurements were used to assess PrISM performances and to compare with state-of-the art precipitation products. This was done to avoid traditionally used products that can be tricky at fine time scales, such as the Global Precipitation Climatology Center (GPCC [26]), the Global Precipitation Climatology Project (GPCP [27,28]) or the Climatic Research Unit (CRU [29]). The paper is organized as follows: Section 2 describes the satellite and ground-based rainfall datasets, and the PrISM algorithm; Section 3 presents the results and a comparison with existing rainfall products; and Section 4 draws conclusions and perspectives.

## 2. Materials and Methods

### 2.1. Ground-Based Precipitation Measurements

Five reference datasets based on in situ rainfall measurements are used in this study (Table 1). The first two datasets are provided by the AMMA-CATCH (Analyse Multidisciplinaire de la Mousson Africaine-Couplage de l'Atmosphère Tropicale et du Cycle Hydrologique) Observatory in Niger and Benin [30,31]. Both sites cover about $1 \times 1°$ and are composed of 34 gauge stations in Niger and 30 gauge stations in Benin. A spatial interpolation (block kriging) was performed at the $0.25° \times 0.25°$ spatial resolution in order to obtain a reference rainfall amount at the commonly used satellite spatial scale (0.25°). The selected 0.25° sites in Niger and Benin are respectively centered at 2.625° E; 13.625° N and 1.625° E; 9.625° N (see Figure 1). The number of gauge stations that directly affect the rainfall amount at the 0.25° resolution are 12 in Niger and 10 in Benin. The third dataset is composed of 20 in situ gauge stations covering the whole of Burkina Faso. The dataset was provided by the National Meteorological Department of Burkina Faso. The fourth dataset is the "WaTFor" dataset which documents Western Central Africa. Built-up by [5], it contains monthly and daily in situ rainfall data collected from global datasets, national meteorological services and monitoring projects for Cameroon, Gabon, Congo and Central African Republic. Finally, the last dataset is a gauge network composed of 78 stations covering seven countries in East Africa (Ethiopia, Djibouti, Somalia, Kenya, Uganda, Tanzania and Rwanda [32]). The datasets at the national scale comprise both synoptic stations (whose data are generally incorporated in post-processed products) and independent stations.

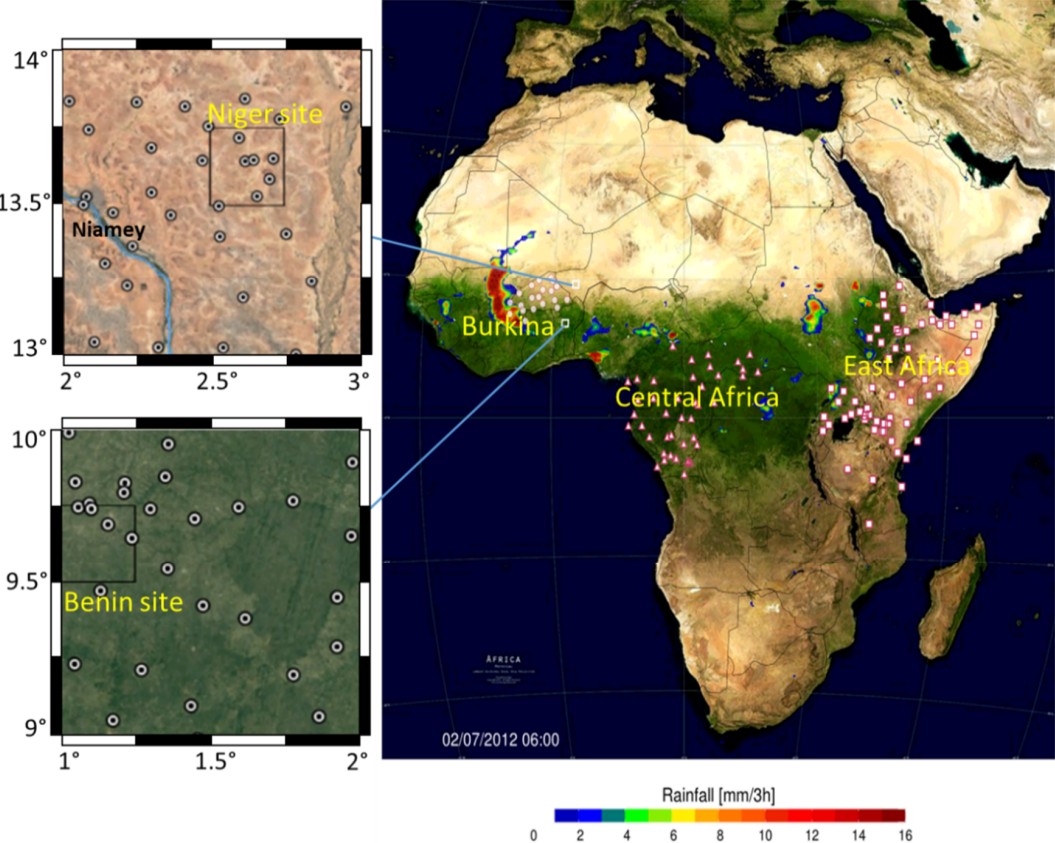

**Figure 1.** Location of the five reference, ground-based rainfall datasets: Burkina Faso (20 stations), Central Africa (42 stations), East Africa (78 stations) and the two AMMA-CATCH sites located in Niger (region of Niamey) and Benin (region of Nalohou). The two squares inside the two left graphs represent the two selected 0.25° pixels in Niger and Benin. Precipitation product on top of land cover on the right map illustrates a time step (3 h) of the PrISM precipitation product (2 July 2012 3 to 6 a.m.).

**Table 1.** Observed rainfall datasets used for evaluation.

| Data Set | Nb Stations | Period | Time Scale |
|---|---|---|---|
| Niger | 12 | 2010–2016 | 3 h |
| Benin | 10 | 2010–2016 | 3 h |
| Burkina Faso | 20 | 2010–2015 | Daily |
| Central Africa | 42 | 2010–2016 | Daily |
| East Africa | 78 | 2010–2013 | Daily |

*2.2. Satellite Precipitation Products*

In addition to the PrISM product, ten existing satellite and ground-based rainfall products have been selected for the inter-comparison (Table 2). All products are provided at the 3 h and 0.25 degree resolution except TAMSAT (Tropical Applications of Meteorology using SATellite data and ground-based observations, daily, 0.0375°), IMERG (Integrated Multi-satellitE Retrievals for GPM, half-hourly, 0.1°), SM2RAIN (Soil Moisture to RAINfall, daily, 0.25°), CHIRPS (Climate Hazards group InfraRed Precipitation with Station data, daily, 0.25°) and GPCC (Global Precipitation Climatology Centre, daily, 1°). In order to enable an inter-comparison of the products, all the products were regridded to 0.25° spatial resolution and daily temporal resolution. Five rainfall products (out of 10) are available in real-time and are not merged with any raingauge observations. The other five products are available after a latency ranging from 7 days to 2 months

**Table 2.** Inventory of all satellite rainfall products' datasets used in the study. To enable the fair comparison of all products, each product was regridded to the 0.25° spatial resolution and daily time-scale.

| Data Set | Spatial Resolution | Time-Scale | Period | Latency | Ground Calibration |
|---|---|---|---|---|---|
| PrISM | 0.25° | 3-hourly | 2010–present | ~5 day | no |
| CMORPH-Raw | 0.25° | 3-hourly | 1998–present | 18 h | no |
| TRMM-RT | 0.25° | 3-hourly | 1998–present | ~6 h | no |
| IMERG-Early | 0.1° | 30 min | March 2015–present | ~12 h | no |
| TAMSAT-v3.0 | 0.0375° | Daily | 1983–present | ~2 days | no |
| SM2RAIN | 0.25° | Daily | 2015–2018 | ~5 days | no |
| CHIRPS-v2.0 | 0.25° | Daily | 1981–present | ~3 weeks | yes |
| GPCC | 1° | Daily | 2009–present | 15–45 days | yes |
| CMORPH-Adj | 0.25° | 3-hourly | 1998–present | >1 month | yes |
| TRMM-3B42 | 0.25° | 3-hourly | 1998–present | >1 month | yes |
| IMERG-Final | 0.1° | 30 min | March 2015–present | >1 month | yes |

CMORPH (CPC Morphing Technique) [33] is a rainfall estimate product based on geostationary infrared images and passive microwave data from low earth-orbiting satellites. Motion vectors are determined from half-hourly geostationary infrared measurements, and used to propagate the estimates obtained from the microwave data. The CMORPH rainfall product is proposed in two modes: the CMORPH-Raw product which is not merged with any raingauge observations and the CMORPH-Adj product which use a monthly ground calibration procedure to remove bias and uncertainties. In this study, the 0.25° 3 h version of the product was used.

TRMM 3B42 products' [34] originality lies in its incorporating precipitation radar and microwave measurements to better evaluate rainfall intensity. It combines these data with polar-orbiting and geostationary satellite images to obtain 3-hourly rainfall estimates at 0.25° spatial resolution. Similar to CMORPH, the TRMM 3B42 products are available in two modes: the TRMM-3B42RT product (hereafter called TRMM-Raw) available in near real-time (latency of about 7 h) which is not merged with any raingauge observations, and the TRMM-3B42 product (hereafter called TRMM-Adj) which includes GPCP monthly gauge aggregations and is available with a latency up to 6 weeks. The TRMM-3B42 v7 used in this study improves upon the previous ones by incorporating additional microwave and

infrared data, revising the relationship between radar reflectivity and rainfall rates, and using better reference databases for bias correction.

The IMERG product, first released in early 2015 [35], is provided at $0.1° \times 0.1°$ spatial and half-hourly temporal resolutions in three modes, based on latency and accuracy: "early" (latency of 4–6 h after observation), "late" (12–18 h), and "final" (~3 months). The main difference between the early and final runs is—beyond the way the different sensor measurements are propagated in time—that the early run has a climatological rain gauge adjustment, while the final run uses a month-to-month adjustment based on GPCC gauge data. In this study, we used the IMERG-Early and IMERG-Final products. The two products were upscaled to 0.25° by using a box-shaped kernel with antialiasing, which was found to outperform simple spatial averaging, and half-hourly rainfall were accumulated to obtain a daily product [36].

Tropical Applications of Meteorology using SATellite and ground-based observations (TAMSAT) was developed at the University of Reading specifically for Africa, with a spatial resolution of 0.0375°. The TAMSAT method [37–39] is based on high resolution (0.0375°) METEOSAT thermal-infrared observations for all of Africa, available from 1983 to the present and updated in near-real time (up to 7 days). Contrary to other merged products, TAMSAT does not use Global Telecommunication System (GTS) data but historical data from about 4000 stations acquired by various African agencies since the early 1990s [40]. We used the TAMSAT V3.0 version product [38] available at the daily timescale. Similar to IMERG, the product was regridded to the 0.25° spatial resolution, and is called TAMSAT_025.

The GPCC daily product is provided by the Global Precipitation Climatology Center [26], and has been available since 1 January 2009 with a spatial sampling grid of 1°. GPCC is a gridded gauge-analysis product derived from quality-controlled station data (more than 85,000 different stations). This dataset is characterized by an uneven spatial distribution: some regions are characterized by dense rain gauge networks (Europe, US, China) while other regions, such as Africa, Amazonia and Northern areas, suffer from low density networks. To enable comparison with other products, GPCC was downscaled to 0.25° spatial resolution with a linear interpolation method.

Climate Hazards group Infrared Precipitation with Stations (CHIRPS) dataset is a quasi-global (50° S–50° N), high-resolution (0.05°), daily, pentadal, and monthly precipitation dataset [41]. CHIRPS uses the Tropical Rainfall Measuring Mission Multi-satellite Precipitation Analysis version 7 (TMPA 3B42 v7) to calibrate global Cold Cloud Duration (CCD) rainfall estimates. CHIRPS incorporates station data in a two phase process. In the first phase, Meteorological Organization's Global Telecommunication System (GTS) gauge data are incorporated and a 2 day latency product is available. In the second phase, station data are combined with monthly (and pentadal) high-resolution rainfall estimates to produce a second product with a latency of about 3 weeks. The version used in this study is the second product, available at the daily and 0.25 degree resolution.

Similar to PrISM, SM2RAIN precipitation product [18,19] takes advantage of satellite soil moisture observations to derive a precipitation product. SM2RAIN is based on the inversion of the soil–water balance equation and allows the estimation of the amount of water entering the soil by using soil moisture observations from in situ or satellite sensors as an input (e.g., [20,42–45]. The SM2RAIN product used in this study is the GPM+SM2RAIN precipitation dataset (hereafter called SM2RAIN) which is based on the integration of IMERG-ER with SM2RAIN-based rainfall estimates derived from ASCAT, SMOS and SMAP L3 soil moisture products. The merging methodology uses an Optimal Linear Combination approach, OLI [46,47]. This approach provides an analytically optimal linear combination of ensemble members (precipitation products, in this case) that minimize mean square error when compared to a reference dataset. The dataset is currently available for Africa and South America (2015–2018), Europe, India, Contiguous United States and Australia (2015–2017) and can be downloaded at https://doi.org/10.5281/zenodo.3345323.

### 2.3. The SMOS Soil Moisture Dataset

The Soil Moisture and Ocean Salinity [48,49] satellite was launched in November 2009 and started delivering data on January 2010. The primary goals of this Earth Explorer mission are to globally and frequently measure surface soil moisture over land, and sea surface salinity over the oceans. The SMOS data used in the study as the main input to the PrISM algorithm correspond to the CATDS level-3 soil moisture data obtained through https://www.catds.fr site. The SMOS-L3SM products are in NetCDF format and were regridded from the EASE 25 km grid to the 0.25° × 0.25° regular grid retained in the present study using the closest neighbor.

### 2.4. The PrISM Methodology

The concept of the Precipitation Inferred from Soil Moisture (PrISM) methodology is to exploit remote sensing soil moisture measurements to correct the amount of rainfall estimated by an existing satellite rainfall product (CMORPH-Raw in this study). It makes use of a simple soil moisture/precipitation model and an assimilation scheme.

#### 2.4.1. The API Soil Moisture/Precipitation Model

The Antecedent Precipitation Index (API) model is a simple model designed to simulate a soil moisture dynamic based on precipitation data. The API model is defined as

$$API_{(t)} = API_{(t-1)} \cdot e^{\frac{\Delta t}{\tau}} + P_{(t)} \tag{1}$$

where $P_{(t)}$ is the rainfall accumulation (in mm) during the period $\Delta t$ (in h), and $\tau$ a parameter that describes the drying-out soil moisture velocity (in h). The *API* index is a simple proxy of the soil moisture dynamic (in mm). Recently, ref. [50] proposed a slight modification of the original API model in order to improve its accuracy and enable the calculation of volumetric soil moisture in $m^3/m^3$ instead of an index expressed in mm. The new version of the API model contains two modifications: (i) it accounts for the degree of saturation of the soil before a rain event; and (ii) the soil moisture is now limited by the saturation value. These modifications to the relationship add three parameters: $d_{soil}$, an equivalent soil thickness (in mm), $\theta_{sat}$, the soil moisture value at saturation (in $m^3/m^3$) and $\theta_{res}$, the residual soil moisture (in $m^3/m^3$). The new version of the API model (hereafter referred as the API model) is written as

$$\theta_{(t)} = \left(\theta_{(t-1)} - \theta_{res}\right) \cdot e^{-\frac{\Delta t}{\tau}} + \left(\theta_{sat} - \left(\theta_{(t-1)} - \theta_{res}\right)\right) \cdot \left(1 - e^{\frac{-P(t)}{dsoil}}\right) + \theta_{res} \tag{2}$$

where $\theta_{(t)}$ is the surface soil moisture in $m^3/m^3$, $\tau$ is the soil moisture drying-out velocity (in h), and $P_{(t)}$ is the cumulative precipitation in mm during the $\Delta t$ period (in h). It requires the use of a precipitation product at infra-daily resolution (3 h or less) to determine when rainfall occurs compared to SMOS ascending (6 a.m.) or descending (6 p.m.) orbits. A sensitivity study was conducted over 10 sites at the global scale [24] to derive the best four parameters of the API model. The authors showed that a constant value for $\theta_{sat}$ = 0.45 $m^3/m^3$ provided reliable results. On the contrary, a spatial distribution of the $\theta_{res}$ and $d_{soil}$ parameters is required as well as a spatiotemporal distribution of the $\tau$ parameter.

The residual soil moisture is the minimal value of soil moisture on a given pixel. Based on surface soil moisture measurements obtained over the 10 sites presented in [24], the simple following formulation was proposed

$$\theta_{res} = 0.04676 + 0.05936 \left(\overline{NDVI}\right) - 0.00136 \left(\overline{Tair}\right) \tag{3}$$

where $\overline{Tair}$ (in °C) is the annual mean 2 m air temperature (source MERRA, 2013) and $\overline{NDVI}$ is the annual mean NDVI value provided by ESA-CCI-LC-L4-NDVI (Spot VGT, 2015). Globally, residual soil

moisture values range from 0.017 to 0.099 m$^3$/m$^3$ at the global scale and from 0.017 to 0.060 m$^3$/m$^3$ in Africa.

The $d_{soil}$ coefficient (in mm) describes the rapidity of soil moisture wetting during a rainfall event and is related to the soil thickness. The thinner (thicker) the soil layer, the faster (slower) the soil wetting. Over nine out of the 10 sites studied in [24], it was found that a $d_{soil}$ value of 35 mm was adequate compared to the in situ soil moisture dynamic. This is consistent with soil moisture depth sensors located at 5 cm depth. However, on the Niger site, a value of $d_{soil}$ equal to 100 mm was required to reproduce observed in situ soil moisture dynamics. It was concluded that this parameter can be related to the presence/absence of vegetation. In regions without vegetation, soils are often degraded, with an impermeable crust associated with a low infiltration rate. A simple sigmoid relationship based on mean annual NDVI (ESA-CCI-LC-L4-NDVI, 2015) was proposed in this study as

$$d_{soil} = 120 - \frac{80}{1 + 178482301e^{(-100*\overline{NDVI})}} \tag{4}$$

Globally, $d_{soil}$ values range from 40 mm (almost everywhere) to 120 mm in arid and semi-arid areas.

The $\tau$ parameter in Equation (2) describes the drying-out velocity of the surface soil moisture due to both evapotranspiration and infiltration rate. Consequently, this parameter should depend on both soil hydraulic properties and atmospheric forcing (air temperature, wind velocity, solar radiation). In a first approximation, it was shown in this study that the $\tau$ value can be appropriately estimated with 30 days smoothed air temperature (*Tair*) using the following relationship

$$\tau(t) = 400 - \left( \frac{350}{\left(1 + e^{-0.1(Tair-7.5)}\right)} \right) \tag{5}$$

where 30 days smoothed *Tair* values (°C) are obtained from MERRA-2 database (3 h, 2013). At the continental scale of Africa, the $\tau$ parameter ranges from 80 h to 350 h.

### 2.4.2. The CDF Matching Procedure

The PrISM methodology is based on the assimilation of the SMOS soil moisture retrievals into the API model (Equation (2)). Thus, a preliminary work consists of scaling the SMOS soil moisture retrievals to the API simulation using a simple CDF-matching procedure. To that end, a reference rainfall dataset was selected to provide a reference soil moisture simulation with the API model at the Africa scale. The evaluation of two adjusted products (CMORPH-adj and TRMM-3B42) against in situ rainfall measurements in Niger and Benin sites led to the selection of the CMORPH-Adj precipitation product as the reference for the CDF-Matching procedure. Then, the API model (Equation (2)) was run for the whole of Africa using CMORPH-Adj precipitation product and parameters were derived from Equations (3)–(5). Based on the obtained reference soil moisture simulation (2012), a calculation of the two linear CDF-matching coefficients (p1 and p2) was made to scale the SMOS L3SM to the reference soil moisture. The scaled SMOS values ($SMOS_{CDF}$) are assumed to be linearly related to SMOS original values as

$$SMOS_{CDF} = p1 + p2.(SMOS) \tag{6}$$

with $p2 = \frac{\sigma_{SMmodel}}{\sigma_{SMsmos}}$ and $p1 = \overline{SM_{model}} - p2.\left(\overline{SM_{smos}}\right)$.

### 2.4.3. The Particle Filter Assimilation Scheme

Among the various existing assimilation schemes, the Particle Filter (PF) is an original method based on random stochastic perturbations of the precipitation forcing that explicitly simulates the consequences of precipitation uncertainties in the associated soil moisture outputs [51–53]. It is suitable for non-linear models and makes no assumption on the prior and posterior distributions of the model states. This property of the PF makes it more suitable for this study compared to ensemble-based data

assimilation approaches whose optimality and performance depend on the linearity between input and output variables, having Gaussian distributed errors, as, for example, in the Ensemble Kalman Filter [54,55]. For a mathematical or formal description of PF, the reader should refer to [56].

The concept is a pixel-based approach. An illustration of the PF assimilation method is shown in Figure 2 (Niger site, 2015). Once there is a new SMOS soil moisture retrieval on a given pixel, an assimilation window which contains the five last SMOS retrievals is defined. The length, i.e., the number of SMOS retrievals within each assimilated sub-period, was chosen after a sensitivity study (not shown) and represents a compromise between too short periods (giving much weight to individual SMOS uncertainties) and too long periods, which reduce the operational interest of the methodology. Thus, the API model is forced with the real-time satellite rainfall product (CMORPH-Raw in this study) that we aim to correct, and which is represented as red "bars" in Figure 2. The red curve in Figure 2 represents the soil moisture simulated by the model forced with this real-time satellite rainfall. Then, the real-time satellite rainfall is used to generate 100 random rainfall time series (number tested in sensitivity experiments) using random stochastic perturbations (grey "bars" in Figure 2). These new rainfall time series are used to force the API model (Equation (2)) to obtain an ensemble of soil moisture predictions (i.e., 100 soil moisture time series associated to 100 different rainfall time series, grey curves in Figure 2). The random stochastic perturbations of rainfall is done using the following simple multiplicative relationship: Rain(t,i) = Rain(t)*a(i), with a(i) a random number between 0 and 2 for i = 1, 100 (uniform distribution). Figure 2 clearly shows that the random rainfall can't exceed twice the amount of the initial rainfall. Then, the SMOS retrievals are used to select the 30 most probable soil moisture trajectories which minimize the RMSE. Finally, the corrected rainfall amount corresponds to the average of the 30 most probable rainfall time-series. The corrected rainfall estimate is associated with an uncertainty calculated as the difference between the maximum and the minimum value of the 30 most probable rainfall time-series.

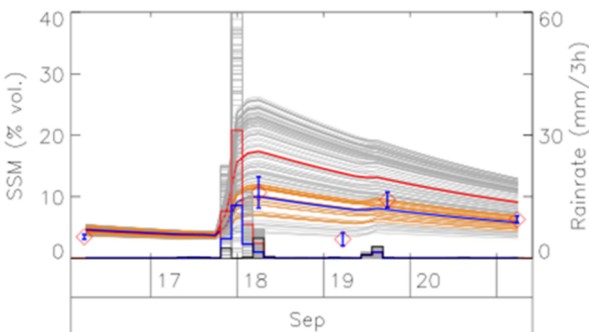

**Figure 2.** Illustration of the PF assimilation scheme for the Niger site. The initial satellite precipitation rate (in red) produces the associated soil moisture evolution (in red). Stochastic perturbations of the initial satellite precipitation rate produce an ensemble of potential soil moisture evolutions (in grey). The SMOS retrievals (five orange diamonds) are used to select the most probable soil moisture curves (in orange) and to calculate the averaged soil moisture (in blue), which is associated with a specific precipitation rate (in blue). In this case, a decrease in the initial satellite precipitation rate is proposed which is consistent with in situ precipitation measurements (in black).

The process is repeated when a new SMOS measurement is available. Consequently, each rainfall event is considered four times (a period of five successive SMOS measurements provides four intervals) and the final rainfall rate correction is the averaged value of the four proposed corrected rainfall rates.

As the method is based on a modification of the precipitation rate of an existing product, it is not possible to create any rain event. It is also difficult to completely remove an existing rain event even if it is possible to significantly reduce it. Therefore, the method has a low impact on scores usually used in satellite precipitation product comparisons such as probability of detection (POD) and false-alarm ratio (FAR).

## 3. Results

To enable fair comparison between satellite precipitation products, all products listed in Table 2 were regridded to the 0.25° resolution and daily timescale. At this spatial scale, a direct comparison with a single raingauge station can be distrusted due to the large spatial scale difference. Thus, we conducted a two-step assessment methodology. First, an accurate assessment was performed on two sites where 10 to 12 raingauge stations are located within the same pixel of 0.25° (i.e., about 25 × 25 km$^2$) belonging to the AMMA-CATCH Observatory in Niger and Benin. As stated in Section 2.1, the use of dense networks of rain gauges allows an accurate estimate of the precipitation rate at this scale. In a second step, a direct, less relevant comparison between satellite (0.25°) and individual rainfall station was done, and results were analyzed at the network scale, i.e., Burkina Faso (20 stations), Central Africa (42 stations) and East Africa (78 stations).

### 3.1. Assessment at the Local Scale (Niger and Benin)

The eleven selected precipitation products were compared to ground-based precipitation measurements using commonly used statistical scores: the Pearson correlation (R), the Root Mean Square Error (RMSE in mm/day) and the annual bias (in mm). Null values were accounted for in the scores calculation and the comparison was performed at the daily time scale and at the 0.25° spatial resolution. As the ground-based precipitation datasets (Niger and Benin) are available at the hourly time scale, whereas some satellite products are provided at the daily time scale, the matching of the two time-series was carefully checked to avoid the known ambiguity between a 6 a.m.–6 a.m. day (commonly used in precipitation measurements) and a 0–24 h day. Lastly, we also examine the number of rainy days (cumulative daily rainfall > 1 mm) compared to in situ measurements.

Results are presented for illustrative purposes for the Benin site (2015) in Figure 3. Statistical scores (R, RMSE and annual bias) are plotted in each graph and are also reported in Table 3. Overall, all products capture relatively well the temporal dynamics of precipitation, with a rather high correlation coefficient (R > 0.70, except for TRMM-Raw (R = 0.65) and GPCC (R = 0.42), reported only in Table 3). The best performances in term of correlation were obtained by PrIMS (R = 0.81), CMORPH-Adj, IMERG-Final (R = 0.80), IMERG-Early (R = 0.78) and CHIRPS (R = 0.77). Regarding RMSE, the best performances were obtained by CMORPH-Adj, PrISM, IMERG-Final and SM2RAIN and IMERG-Early, with RMSE values equal to 4.3, 4.4, 4.5 and 4.6 mm/day, respectively. The lowest performances were obtained by TRMM-Raw and GPCC, with 6.8 mm/day. Regarding bias score, four products obtained annual cumulative precipitation values very close to the observation (1150 mm): TRMM-Raw (1138 mm, −1%), PrISM (1124 mm, −2.3%), CHIRPS (1110 mm, −4.3%) and SM2RAIN (1216 mm, +5.7%). On the contrary, TAMSAT_025 strongly underestimated the annual precipitation with an estimation of only 731 mm (−36%). Surprisingly, the three adjusted products (CMORPH-Adj, TRMM-Adj and IMERG-Final) provided moderate to strongly underestimated precipitation (respectively 828 mm (−28%), 1075 mm (−6.5%) and 1035 mm (−10%) compared to in situ precipitation measurements. Similarly, GPCC exhibited moderate underestimation (1013 mm, −12%). Only the CMORPH-Raw product showed an overestimation of the annual rainfall, with 1246 mm (+8%) in Benin in 2015. Lastly, Table 3 includes the difference in terms of annual rainy days. In the Benin site, 103 rainy days (>1 mm) were observed in 2015. Table 3 shows that CHIRPS, GPCC and SM2RAIN tend to overestimate this number (112, 135 and 168 days respectively), whereas PrISM and CMORPH-Raw provided a similar number of rainy days (101 and 102 days respectively). All other precipitation products exhibited a slight underestimation (from 88 to 94 days).

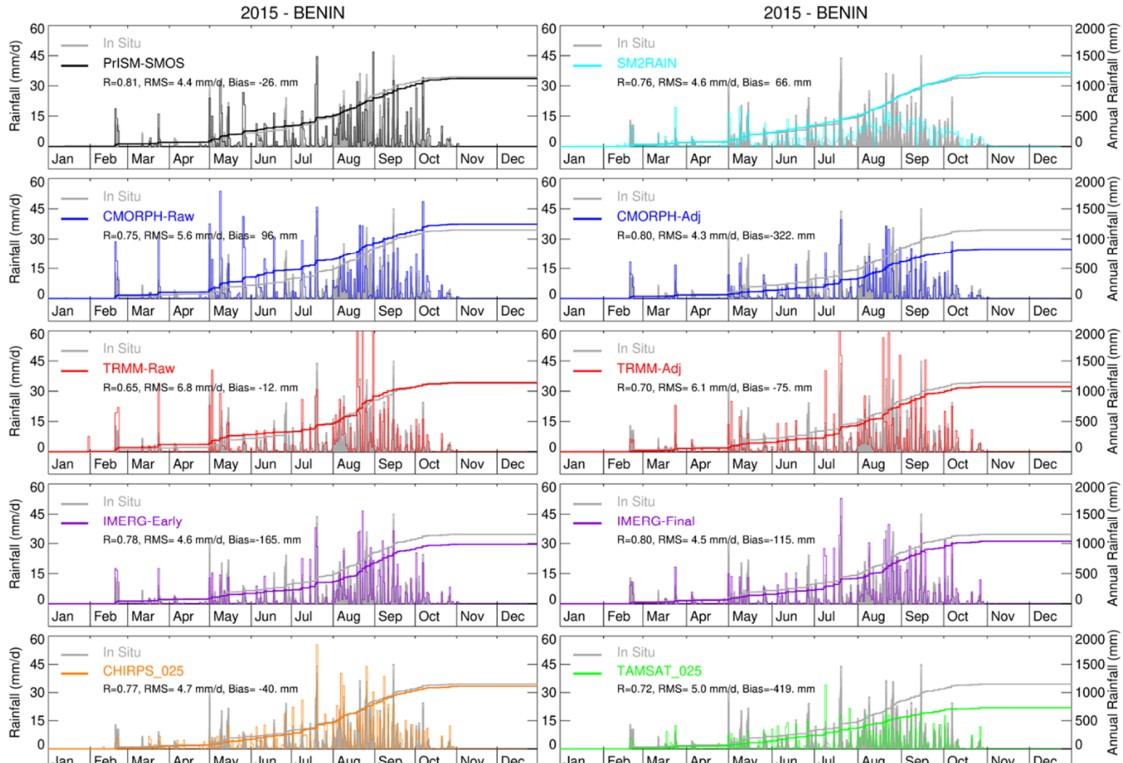

**Figure 3.** Example of comparison between in situ precipitation measurements (Benin 0.25° site, 2015, in grey) and the ten precipitation products (Precipitation Inferred from Soil Moisture (PrISM), SM2RAIN, CMORPH (Raw and Adj), TRMM (Raw and Adj), IMERG (Early and Final), Climate Hazards group Infrared Precipitation with Stations (CHIRPS)-025 and Tropical Applications of Meteorology using SATellite and ground-based operations (AMSAT)-025). Bars show daily rainfall amounts (left axis), curves show cumulative rainfall (right axis). Statistical scores are reported in Table 3.

**Table 3.** Statistical scores (R, RMSE, annual bias and nb of rainy days) between in situ precipitations measurements (Benin and Niger, 2015) and the eleven precipitation products. Bold values indicate the best performances. There are 103 rainy days in Benin and 43 rainy days in Niger in 2015.

| 2015 | Benin 0.25° (1150 mm, 103 Rainy Days) | | | | Niger 0.25° (601 mm, 43 Rainy Days) | | | |
|---|---|---|---|---|---|---|---|---|
| | **R** | **RMSE (mm/d)** | **Bias (mm)** | **Rainy Days (>1 mm/d)** | **R** | **RMSE (mm/d)** | **Bias (mm)** | **Rainy Days (>1 mm/d)** |
| PrISM | **0.81** | **4.4** | **−26** | **101** | **0.81** | **3.7** | **+68** | 55 |
| CMORPH-Raw | 0.75 | 5.6 | +96 | **102** | **0.80** | 6.7 | +451 | 56 |
| TRMM-Raw | 0.65 | 6.8 | **−12** | 94 | 0.75 | 6.9 | +489 | 54 |
| IMERG-Early | **0.78** | **4.6** | −165 | 90 | 0.63 | 5.3 | **+86** | 57 |
| TAMSAT-025 | 0.72 | 5.0 | −419 | 92 | 0.77 | **3.9** | −169 | **50** |
| SM2RAIN | **0.76** | **4.6** | **+66** | 168 | 0.74 | **4.1** | −203 | **48** |
| CHIRPS | **0.77** | **4.7** | **−40** | 112 | 0.70 | **4.3** | −138 | 55 |
| GPCC | 0.42 | 6.8 | −137 | 135 | 0.33 | 5.9 | −147 | 71 |
| CMORPH-Adj | **0.80** | **4.3** | −322 | 88 | **0.82** | **4.3** | +152 | **51** |
| TRMM-Adj | 0.70 | 6.1 | **−75** | 92 | 0.75 | **4.1** | **+1** | 53 |
| IMERG-Final | **0.80** | **4.5** | −115 | 94 | 0.66 | **4.6** | **−54** | 52 |

The same analysis was conducted for the Niger site in 2015. Graphics are shown in Appendix A (Figure A1) and the statistical scores are reported in Table 3. Similarly, the CMORPH-Adj and PrISM products obtain good performances in term of correlation (R = 0.82 and R = 0.81, respectively) and GPCC obtains the lower score (R = 0.33) probably due to the low density of gauges in the Niger region and its original spatial resolution of 1°. Regarding the RMSE scores, best performances are obtained by PrISM, TAMSAT-025, CHIRPS, TRMM-Adj, SM2RAIN, CMORPH-Adj and IMERG-Final (from 3.7 to

4.6 mm/day). Conversely, CMORPH-Raw and TRMM-Raw products obtain lower RMSE scores (6.7 and 6.9 mm/day). In terms of annual bias, four products obtained annual cumulative precipitation values quite close to the observation (601 mm): TRMM-Adj (602 mm), IMERG-Final (547 mm, −9%), PrISM (669 mm, +11%) and IMERG-Early (687 mm, +14%). On the contrary, CMORPH-Raw and TRMM-Raw strongly overestimate annual precipitation (1052 mm, +75% and 1090 mm, +81%, respectively). Other products obtain intermediate annual bias (see Table 3). Regarding the number of rainy days in Niger (43 days from in situ measurements), best performances were obtained by SM2RAIN, TAMSAT_025 and CMORPH-Adj (48, 50 and 51 days, respectively) whereas GPCC still provides an overestimated number of rainy days (71 days). Other products, PrISM included, provide a slight overestimation of the number of rainy days compared to observations (from 52 to 57 days).

Figure 4 shows the Taylor diagrams [57] of the Benin and Niger sites' daily precipitation in 2015. The Taylor diagram enables a visual comparative assessment of the different precipitation products quantifying the degree of correspondence between the estimated and observed precipitation in terms of three statistics: the Pearson correlation coefficient, the root–mean–square error (RMSE), and the standard deviation. Correlation scores are plotted as the radial lines, and the linear distance of a point to the Observed point indicates the RMSE from in situ measurements. For instance, the GPCC point for the Benin site is close to the 0.4 radial line (R = 0.42 in Table 3) and close to the "7" dotted circle (RMSE = 6.8 mm/days). Graphically, the best products are those closest to the "Observed" point and to the dashed curve which indicates the same amplitude of the variations. In Benin (2015), the best products are the two IMERG products and the PrISM product. For Niger (2015), PrISM and TRMM-Adj perform better than the other ones.

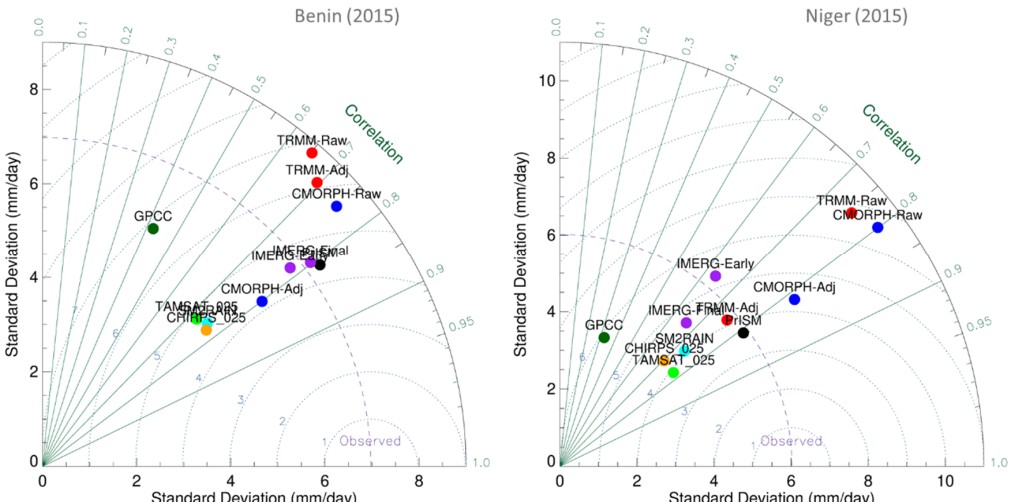

**Figure 4.** Taylor diagrams for the Benin (**left**) and Niger (**right**) 0.25° sites for the year 2015 for the 11 precipitation products. Color dots refer to the different products (blue = CMORPH, red = TRMM, purple = IMERG, green = TAMSAT_025, cyan = SM2RAIN, orange = CHIRPS, dark green = GPCC and black = PrISM). Statistical scores are given in Table 3.

Lastly, the Taylor diagrams were plotted for the whole period (2010–2016, instead of 2015) except for the two IMERG and SM2RAIN precipitation products, which start in March 2014 and are then considered only for 2015–2016 period. Figure 5 reveals some slight differences compared to Figure 4 but leads to similar conclusions. Globally, when a product is provided in two versions, e.g., raw and adjusted, the adjusted product performs better. The "raw" version of TRMM and CMORPH are located far away from their adjusted versions. The PrISM product was found to be among the best products for the Benin site and provides the best performance for the Niger site. TAMSAT_025 and SM2RAIN have similar scores and perform better for the Niger site but their standard deviations are much lower than in the observation.

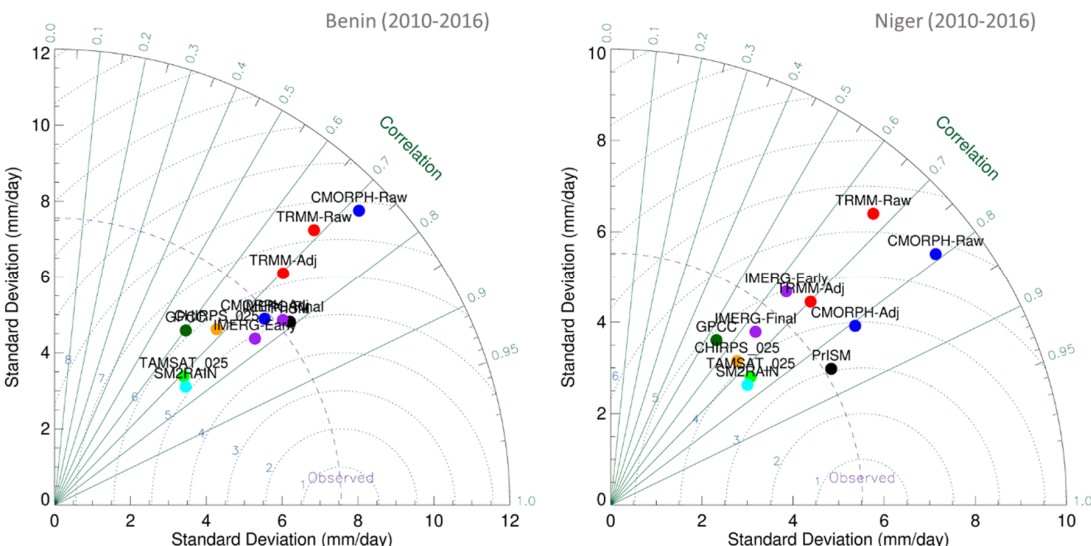

**Figure 5.** Taylor diagrams for the Benin (**left**) and Niger (**right**) 0.25° sites and the 2010–2016 period (except for IMERG, 2015–2016) and the 11 precipitation products. Color dots refer to the different products (blue = CMORPH, red = TRMM, purple = IMERG, green = TAMSAT_025, cyan = SM2RAIN, orange = GPCC and black = PrISM).

## 3.2. Assessment at the Regional Scale

At the regional scale, scores were calculated using individual rain-gauge stations compared to 0.25° precipitation products. Scores were calculated at the daily time scale based on the whole available period (see Table 1). Then, the median value of all individual scores (R, RMSE and bias) was calculated. Results are presented separately for the three rain-gauge networks (Burkina Faso, Central Africa and East Africa) in Figure 6. Results obtained for Benin and Niger are also reported in Figure 6 for comparison. Note that the two IMERG products were not considered for East Africa because the ground rainfall measurements were not available after 2013.

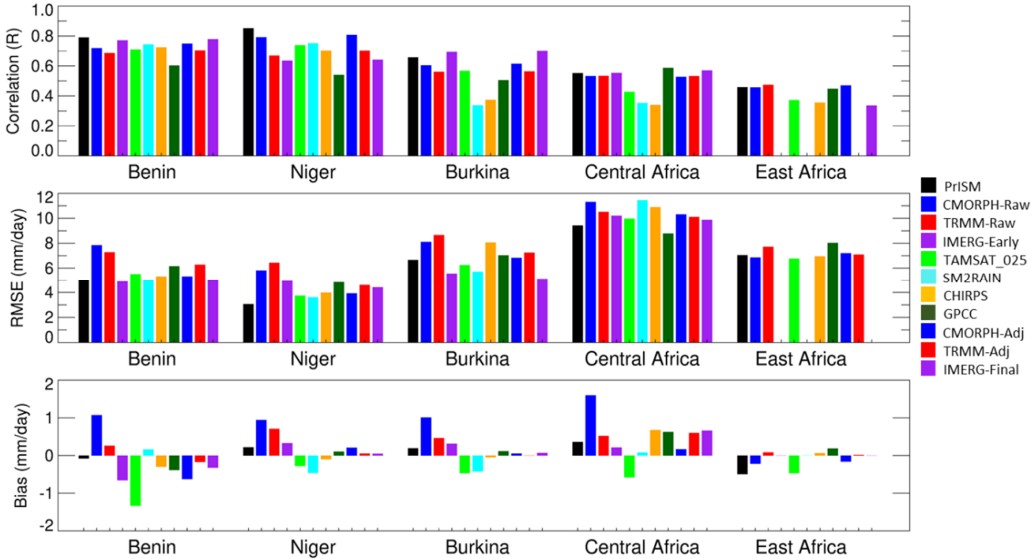

**Figure 6.** Statistical scores (R, root–mean–square error (RMSE) and bias) for the five regions (Benin, Niger, Burkina-Faso, Central Africa and East Africa) and for the 11 precipitation products. The temporal period depends on the satellite product and in situ availabilities.

Results show, globally, that the correlation score is slightly lower for Burkina, Central Africa and East Africa compared to Benin and Niger. Similarly, the corresponding RMSE are respectively greater (worse) for these three networks. On the other hand, the biases are in the same order of magnitude in the five regions. In all regions, PrISM shows good performances according to the three scores and is among the best products, together with the two IMERG products. It can be observed that PrISM systematically outperforms CMORPH-Raw in term of RMSE (except in East Africa, where it is slightly less performant). However, the improvement in terms of correlation is more modest. CMORPH and TRMM, in their adjusted or raw versions, show similar performances in terms of correlation but generally provide much better results in terms of RMSE and biases for their adjusted versions. SM2RAIN and CHIRPS products show a lower correlation and higher RMSE scores for Burkina-Faso and Central Africa but present a low bias for all regions. Although its biases are small, GPCC, as expected from its low original resolution, generally shows relatively low performances, except in Central Africa. Finally, TAMSAT_025 globally tends to underestimate the rainfall rate but shows relatively good correlation and RMSE scores.

## 4. Limitation of the PrISM Methodology

One limitation of the methodology is that it is not able to create a rain event. This is clearly shown in Table 3, where the number of rainy days (>1 mm) does not change much between the CMORPH-Raw product and the PrISM product. Therefore, it is necessary to use an initial precipitation product that overestimates the number of events, since the PrISM algorithm is able to reduce the amount of rainfall.

The PrISM algorithm was also found to be more efficient when the initial rainfall product overestimates the annual rainfall amount. This is particularly true in East Africa, where the CMORPH-Raw showed a negative annual bias (see Figure 6) and the PrISM algorithm was not able to correct for that underestimation. On the contrary, in the other four sites, the CMORPH-Raw showed positive annual biases which are suitably corrected by PrISM. The reason for this behavior is that a 10 mm rain event (for instance) can easily be reduced to 1 mm, but the PrISM algorithm can't propose a correction greater than 20 mm (correction factor ranging between 0 and 2). Consequently, it is easier to reduce the rainfall amount than to increase it.

The satisfactory results of the PrISM methodology in Central Africa are a pleasant surprise, since the SMOS soil moisture retrievals under dense forest are expected to be inaccurate. These results are partly due to the large overestimation of the CMORPH-Raw product in Central Africa. The PrISM methodology reduces the rainfall amount of most events and, mechanically, RMSE is reduced, as well as the annual bias. Overall, the effect of vegetation cover on the performance of the PrISM algorithm is difficult to evaluate. Indeed, areas of dense vegetation are areas where the SMOS signal can be inaccurate, but they are also areas of heavy precipitation with a high potential for improvement. PrISM performances are similar in Niger (low vegetation) and in Benin (medium vegetation), and the PrISM performances in Central Africa are better than in East Africa (in terms of correlation and bias correction).

## 5. Summary and Next Step

This study presents the PrISM algorithm and its evaluation over five regions in Africa. The PrISM algorithm uses knowledge of soil water content (provided by SMOS soil moisture measurements) to adjust the precipitation rate of an existing satellite product (CMORPH-raw in this study). To assess the benefit of the proposed methodology, PrISM was compared against ten state-of-the-art satellite and ground-based rainfall products for five rain-gauge networks located in semi-arid to wet areas in Africa.

PrISM was found to generally outperform all real-time products (CMORPH-Raw, TRMM-Raw, IMERG-Early, TAMSAT_025 and SM2RAIN), especially when considering areas where there is a dense network of rain-gauge stations as a reference dataset. It showed the same or even better performances than adjusted or post-processed products. The main contribution of PrISM is that it greatly decreases RMSE values and reduces the bias values compared with the original CMORPH-Raw product. Results

in term of correlations are more modest. This result is quite important for many applications that require real-time information on precipitation, such as crop yield estimates, flood nowcasting, dam management, groundwater recharge estimates and irrigation demand over large areas.

Future studies will be designed to apply the PrISM algorithm to other satellite soil moisture datasets such as SMAP, ASCAT or SMOS-IC. Currently, the PrISM product is available on the ftp site: ftp://ftp.ifremer.fr/Land_products/L4_PrISM/Africa/ and can be downloaded at https://doi.org/10.5281/zenodo.3565610 at an annual latency. The near-real-time version of the product will shortly be available on the external ERDDAP of IGE in Grenoble: http://osug-smos-rea.osug.fr:8081/erddap/index.html.

**Author Contributions:** T.P. proposed the idea, carried out the experimental design, analyzed the data, and wrote the paper. C.R.-C. and B.P. participated in the PrIMS algorithm development and validation, C.B., P.C., N.P. and G.P. provided in situ rainfall data; G.Q. and R.B. helped in the visualization, L.B. and C.M. provide SM2RAIN product and contribute to the writing of the paper. Y.H.K. contributed to the original idea and to the writing of the paper. D.F.-P. was the coordinator of the SMOS + RAINFALL ESA project. All authors have read and agreed to the published version of the manuscript.

**Funding:** This research was funded by ESA, grant number ESA/AO/1-7875/14/I-NC (4000114738/15/I-SBO). This research also benefits from TOSCA (CNES, Centre national d'études spatiales, France).

**Acknowledgments:** The authors would like to thank University of Douala, University of Bangui, University Omar-Bongo in Libreville, University of Liège, CIRAD Montpellier for providing precipitation data in Central Africa, and National Meteorological Department of Burkina Faso.

**Conflicts of Interest:** The authors declare no conflict of interest.

## Appendix A

Similar to Figure 3, Figure A1 shows the performances of ten precipitation products compared to in situ measurements in the Niger site (2015).

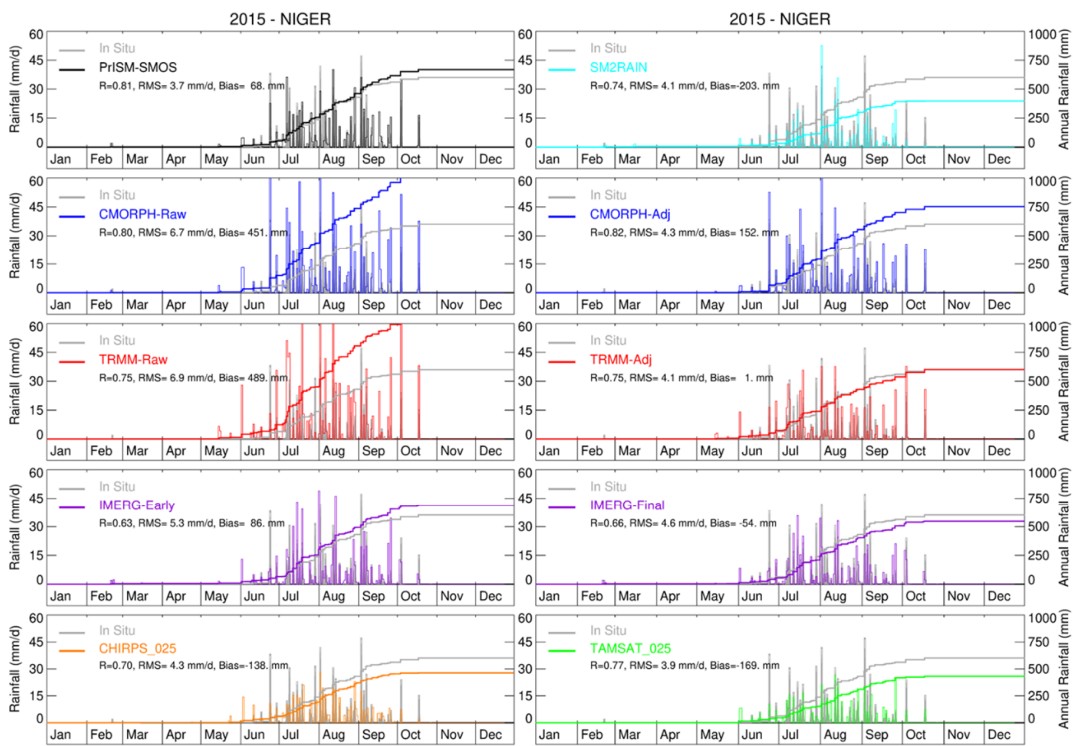

**Figure A1.** Example of comparison between in situ precipitation measurements (Niger 0.25° site, 2015, in grey) and the ten precipitation products (PrISM, SM2RAIN, CMORPH (Raw and Adj), TRMM (Raw and Adj), IMERG (Early and Final), CHIRPS-025 and TAMSAT-025). Statistical scores are reported in Table 3.

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
