# Peer review of "The Precipitation Inferred from Soil Moisture (PrISM) Near Real-Time Rainfall Product: Evaluation and Comparison"

_remotesensing, doi:10.3390/rs12030481_

Round 1

Reviewer 1 Report

This study proposes to improve satellite-based precipitation product over Africa by assimilating SMOS retrieved soil moisture, and promising results are obtained which imply future applications. The manuscript is in general well written. I am pleased to recommend this work being published on Remote Sensing with minor revision by taking care of the following comments/concerns.

Specific comments

L33: “… is assessed …” L98: “two first” -> “first two” L105: please finish the sentence: “The dataset was provided by …” L110: I count “7” countries other than “4” Figure 1: better to place dataset names around corresponding stations on the map L124-125: “precipitation product on top of land cover”? L129: “provided to” -> “provided at” L162: may need reference that supporting this “outperformance” L180: “unable” should be “enable” L274: need reference for Eq.5 L317: guess this “real-time satellite rainfall” refer to “CMORPH-Raw” (right?) according to your statement in the conclusion part (L501-503). Meanwhile, it also confuses me that the precipitation datasets used for CDF-matching (in section 2.4.2) and API modeling are not the same. Therefore, I would encourage the authors to: 1) explicitly point out the objective of this study, i.e. to improve the CMORPH-Raw product, at the front of this manuscript (probably around L81-84), and 2) clarify specific utilization of different precipitation datasets in section 2.2. L431: “RMSE=6.8 days” -> “RMSE=6.8 mm/day” L524-527: please substitute this part with “real” acknowledgements, if any.

Author Response

The answers to reviewer#1 are given below (in italic and bold).

L33: “… is assessed …” L98: “two first” -> “first two” -> done

L105: please finish the sentence: “The dataset was provided by …” The dataset was provided by the National Meteorological Department of Burkina Faso

L110: I count “7” countries other than “4” -> done

Figure 1: better to place dataset names around corresponding stations on the map -> done, Figure 1 was modified accordingly

L124-125: “precipitation product on top of land cover”? -> done: Precipitation product on top of land cover on the right map illustrates a time step (3h) of the PrISM precipitation product (2012 July 2nd 3 to 6 am).

L129: “provided to” -> “provided at” -> done

L162: may need reference that supporting this “outperformance” -> the reference was added in the text (Massari et al., 2020, HESSD)

L180: “unable” should be “enable” -> done

L274: need reference for Eq.5 -> This equation was developed in the present study. This was added in the text : “In a first approximation, it was shown in this study that τ value can be appropriately estimated with 30-days smoothed air temperature (Tair)”

L317: guess this “real-time satellite rainfall” refer to “CMORPH-Raw” (right?) according to your statement in the conclusion part (L501-503). Meanwhile, it also confuses me that the precipitation datasets used for CDF-matching (in section 2.4.2) and API modeling are not the same. Therefore, I would encourage the authors to: 1) explicitly point out the objective of this study, i.e. to improve the CMORPH-Raw product, at the front of this manuscript (probably around L81-84), and 2) clarify specific utilization of different precipitation datasets in section 2.2.

Yes, the reviewer is right, the CMORPH-Raw product was used as the initial real-time product in the PrISM methodology. To avoid confusion, this is now clearly indicated in the four lines describing the PrISM methodology (Section 2.4).

L431: “RMSE=6.8 days” -> “RMSE=6.8 mm/day” -> done

L524-527: please substitute this part with “real” acknowledgements, if any. -> done

Reviewer 2 Report

The study by Pellarin et al., 2020 presents a very interesting study on the generation of precipitation product with the aid of satellite based SMOS soil moisture products. The study is well designed and the manuscript is well written and presented. I believe it is relevant for the journal Remote sensing and is recommended for publication after minor revision. Below are my suggestions.

Various satellite based precipitation products were used to provide indirect evaluation of produced precipitation dataset. Why do you choose these products rather than others? What is your selection criteria? As far as I know, Chirps dataset has been validated in Africa and found to be outperform other precipitation products. I am curious how good is your product compared to Chrips?

Regarding to assessment at the regional scale, it might be more informative if you present a spatial pattern of the different products.

Author Response

The answers to reviewer#2 are given below (in italic and bold).

Various satellite based precipitation products were used to provide indirect evaluation of produced precipitation dataset. Why do you choose these products rather than others? What is your selection criteria? As far as I know, Chirps dataset has been validated in Africa and found to be outperform other precipitation products. I am curious how good is your product compared to Chrips?

The selection of the satellite-based precipitation products was designed to get state-of-the-art products (CMORPH, TRMM, IMERG), and usually used product in Africa (TAMSAT). However, we recognise that the CHIRPS product is now widely used as well, and we have decided to add it to the study. Chirps analysis was added to the new version of the paper.

Figures 3, 4, 5 and 6 as well as Table 2 and 3 were modified. The following paragraph was added to describe the CHIRPS dataset : “CHIRPS (Climate Hazards group Infrared Precipitation with Stations) dataset is a quasi-global (50°S-50°N), high resolution (0.05°), daily, pentadal, and monthly precipitation dataset [41]. CHIRPS uses the Tropical Rainfall Measuring Mission Multi-satellite Precipitation Analysis version 7 (TMPA 3B42 v7) to calibrate global Cold Cloud Duration (CCD) rainfall estimates. CHIRPS incorporates station data in a two phase process. In the first phase, Meteorological Organization’s Global Telecommunication System (GTS) gauge data are incorporated and a 2-day latency product is available. In the second phase, station data are combined with monthly (and pentadal) high resolution rainfall estimates to produce a second product with a latency of about 3 weeks. The version used in this study is the second product available at the daily and 0.25 degree resolution.”

Regarding to assessment at the regional scale, it might be more informative if you present a spatial pattern of the different products.

Thank you for the suggestion. The approach of the paper was to assess satellite-based products using ground-based precipitation observations. A 2D visualisation of the eleven products (mean, standard deviation, or rainfall season length,…) can be interesting to compare products with each other but will not provide any information about the “true” precipitation field as it can be obtained using in situ precipitation measurements. Unless the reviewer has a precise idea of the variable to be presented in 2D, it does not seem relevant to us to produce these maps.

Reviewer 3 Report

Overall the manuscript is well written focusing the significance of PrISM algorithm. Introduction part can be improved. Methods and Results section are well written. Good study. Here are few comments:

Line 105: The dataset was provided by …..? Please mention who provided the dataset.

Line 109: Replace 4 with 7 ?? (as you have provided the list of 7 countries)

Line 214: Please describe in more depth of the uncertainties/limitations and other factors that impact the reliability of API model. How extreme weather conditions impact the performance of the model?

Line 403: Material

Please discuss briefly on the roles of other variables (soil types?, canopy cover) etc in estimating precipitation product (combined with satellite and soil moisture)?

Author Response

The answers to reviewer#3 are given below (in italic and bold).

Overall the manuscript is well written focusing the significance of PrISM algorithm. Introduction part can be improved. Methods and Results section are well written. Good study. Here are few comments:

Line 105: The dataset was provided by …..? Please mention who provided the dataset. -> The dataset was provided by the National Meteorological Department of Burkina Faso

Line 109: Replace 4 with 7 ?? (as you have provided the list of 7 countries), Thank you for pointing out that mistake ;-)

Line 214: Please describe in more depth of the uncertainties/limitations and other factors that impact the reliability of API model. How extreme weather conditions impact the performance of the model?

A new section (“Discussion”) was added in the new version of the manuscript. This section discuss about the limitations of the PrISM methodology : 

Limitation of the PrISM methodology

One limitation of the methodology is that it is not able to create a rain event. This is clearly shown in Table 3 where the number of rainy days (> 1 mm) does not change much between the CMORPH-Raw product and the PrISM product. Therefore, it is necessary to use an initial precipitation product that overestimates the number of events since the PrISM algorithm is able to reduce the amount of rainfall.

The PrISM algorithm was also found to be more efficient when the initial rainfall product overestimates the annual rainfall amount. This is particularly true in East Africa where the CMORPH-Raw showed a negative annual bias (see Figure 6) and the PrISM algorithm was not able to correct for that underestimation. On the contrary, on the other 4 sites, the CMORPH-Raw showed positive annual biases which are suitably corrected by PrISM. The reason for this behavior is that a 10 mm rain event (for instance) can easily be reduced to 1 mm, but the PrISM algorithm can’t propose a correction greater than 20 mm (correction factor ranging between 0 and 2). Consequently, it is easier to reduce the rainfall amount than to increase it.

The satisfactory results of the PrISM methodology in Central Africa are a pleasant surprise since the SMOS soil moisture retrievals under dense forest are expected to be inaccurate. The explanation of this results is partly due to the large overestimation of the CMORPH-Raw product in Central Africa. The PrISM methodology leads to reduce the rainfall amount of most events and, mechanically, RMSE is reduced as well as the annual bias.  

Line 403: Material -> done

Please discuss briefly on the roles of other variables (soil types?, canopy cover) etc in estimating precipitation product (combined with satellite and soil moisture)?

The effect of the vegetation cover on the performance of the PrISM algorithm is difficult to evaluate. Indeed, areas of dense vegetation are at the same time areas where the SMOS signal can be inaccurate but also areas of heavy precipitation with a high potential for improvement. PrISM performances are similar in Niger (low vegetation) and in Benin (medium vegetation), and the PrISM performances in Central Africa are better than in East Africa (in term of correlation and bias correction)”. -> This text was added in the new manuscript.